# Intra Complexity Control Algorithm for VVC

**Zhengjie Shu [1], Junyi Li [2], Zongju Peng [1,2,*], Fen Chen [1,2] and Mei Yu [1]**

[1]  Faculty of Information Science and Engineering, Ningbo University, Ningbo 315211, China
[2]  School of Electrical and Electronic Engineering, Chongqing University of Technology, Chongqing 400054, China
*  Correspondence: pengzongju@nbu.edu.cn

**Abstract:** Versatile Video Coding (VVC) has advantages over High Efficiency Video Coding (HEVC); it can save nearly half of the bit rate and significantly improve the compression efficiency, but VVC's coding complexity is extremely high. Therefore, VVC encoders are difficult to implement in video devices with different computing capabilities and power constraints. In this paper, we apply texture information and propose a VVC intra complexity control algorithm. The algorithm assigns a different encoding time to each CU based on the corresponding texture entropy. Besides, the complexity reduction strategy at the CU level is designed to balance the complexity control while taking rate-distortion performance into consideration. Experiments in our paper show that the coding complexity can be accurately controlled from 90% to 70% with a slight loss of RD performance.

**Keywords:** VVC; complexity allocation; complexity control; intra coding; coding unit

## 1. Introduction

In the last decades, the amount of video transmitted through broadcast channels, media platforms, and virtual networks has grown significantly. Demand for HD and ultra HD video, 360 omnidirectional video, and high fps video has increased. The quantity of information to be transmitted and the bandwidth requirements are also greatly increased, which places a huge burden on the communication transmission network. To settle the transmission pressure, the Joint Video Experts Group introduced quantities of new technologies into the coding framework and formulated the Versatile Video Coding (VVC) standard. With the same quality of experience, the VVC encoder has higher compression efficiency in comparison to the current High Efficiency Video Coding (HEVC) standard. Meanwhile, each mobile terminal has different computing capabilities and different video compression speeds. Therefore, different complexity control algorithms need to be specified for different terminals.

The fast coding method is the foundation for the complexity control algorithm. Therefore, to lessen the coding complexity of VVC, Zhang et al. [1] estimate CU's texture direction information with the gray level co-occurrence matrix, and terminated the horizontal or vertical direction partition. Liu et al. [2] utilized CU's cross-block disparity to skip the not optimal partitions. In [3], a Support Vector Machine (SVM) primarily based method is proposed. The approach makes use of texture information to perform early termination. Saldanha et al. [4] skipped partition patterns that are unlikely to be selected as the best partition type by an optical gradient boosting machine classifier. Kulupan et al. [5] accelerated the CU partition decision by selecting appropriate features to better specialize the prediction of block features. Wang et al. [6] terminated the iterations early with a joint classification decision tree. A random forest scheme was also introduced to reduce partition redundancy in [7]. Fu et al. [8] considered that horizontal or vertical partitions will bring a lot of coding complexity and used a Bayesian-based classifier to skip redundant partitions. Yang et al. [9] introduced a fast decision framework for QTMT structure selection and intra-mode search. Meanwhile, heuristic algorithms are able to prune the Quad Tree

with a nested Multi-type Tree (QTMT) structure. Lei et al. [10] proposed a sum of absolute transformed differences to determine possible block sizes. Chen et al. [11] addressed the rectangular partition issue with variance and gradient features. Cui et al. [12] predetermined the likelihood of directional partitions. However, the above methods that use artificial features cannot describe the segmentation rules well, so the coding loss becomes large as the complexity decreases. He et al. [13] divided CUs into three categories, namely simple ones, normal ones, and complex ones. A random forest classifier will be introduced to simple and complex CUs to reduce unnecessary partition recursion. Normal CUs will be predicted whether they will continue to be partitioned.

For neural network based methods, the main problem is the flexibility of the size of the CU as input. Wu et al. [14] used a hierarchical grid fully convolutional network framework combined with parallel processing and a hierarchy grid map to achieve a partition structure. Pan et al. [15] introduced a multi-information fusion convolutional neural network (CNN) model into the coding fast algorithm, which is used to prematurely terminate merge mode decisions. A multi-stage dropout CNN model was proposed by Li et al. [16] to determine CU partitions to conform to the multi-stage flexible QTMT structure. Then, an adaptive loss function is designed to train a multi-stage dropout CNN model, combining the uncertain number of segmentation patterns and the objective of minimizing Rate-Distortion (RD) cost. Lastly, a balance between complexity and RD performance is obtained by a multi-threshold scheme. Park et al. [17] introduced a lightweight neural network to reduce the computational complexity caused by the ternary tree partition in VVC. Zhang et al. [18] proposed DenseNet-based probability prediction. The method obtained a probability for each QTMT partition and for skip partitions by comparing the probability with the threshold. Tech et al. [19] proposed a CNN algorithm for $32 \times 32$ CU blocks that minimizes the rate-distortion function as the objective function. Jin et al. [20] found that depth range and coding complexity are strongly correlated. The method reduced the depth range of each $32 \times 32$ CU to terminate the Rate-Distortion Optimization (RDO). Wang et al. [21] extended depth range prediction to $64 \times 64$ CU for inter coding.

There have been some complexity control algorithms for video coding. Most of the research is on HEVC and there is little on VVC. To control the complexity of HEVC, Huang et al. [22] designed a variable accuracy CU decision model, which can control the coding complexity by changing the model's accuracy. Cai et al. [23] focused on achieving a constant objective reconstruction quality during video encoding, by modeling bitrate and distortion as functions of video constituents and control features to achieve a certain rate-distortion performance. Li et al. [24] found that most computations can be pruned. Then, an adaptive pruning scheme is devised to apply well-suited weight parameter retention rates to each level. Finally, complexity control is accomplished with several network models generated by different retention rates. Huang et al. [25] proposed a heuristic directed framework where the HEVC encoder can be adapted to the underlying acceleration algorithm. In this framework, CU and PU partitions are accelerated by boundary-consideration CNN and Naive Bayes, respectively. Deng et al. [26] combined the subjective experience with complexity control. This research not only discussed how the CU partition depth range relates to subjective distortion, but also additionally managed the coding complexity via restricting the range of CU partition depth. Jimenez-Moreno et al. [27] applied the parameter adjustment to the complexity management method. The research studied the relationship between the CU partition threshold and RDO iteration complexity and used a feedback mechanism to adjust the model threshold. Zhang et al. [28] estimated and controlled the complexity of CTU-level, taking advantage of the flexibility of HEVC. To control the complexity for VVC, Huang et al. [29] proposed a Time-PlanarCost model for CTU-level encoding time estimation and control.

The VVC fast coding method and HEVC complexity control algorithms need to be redesigned for VVC complexity control. The method in [29] controls the maximum depth of $64 \times 64$ without deciding each sub-CU's partition. For more accurate control of CU-level partitions and coding, this paper proposes a complexity control algorithm that assigns

different fast algorithms to each CU according to the texture information entropy. Our experimental results demonstrate that the coding performance is acceptable and the coding complexity can be effectively controlled by the proposed algorithm.

## 2. Methods

### 2.1. Algorithm Framework

Figure 1 illustrates the whole process of our algorithm. The algorithm mainly consists of two parts—the training algorithm and the prediction algorithm. If the current frame is used for model training and data collection, the proposed method performs normal encoding, collects SVM samples, and trains SVM models. The trained SVM CU partition models will be introduced to predict the partition direction of CUs in subsequent frames. If the current frame is a prediction frame, the method will predict the coding complexity of the current frame and use the texture information entropy at the CU-level to assign the coding complexity to each CU. Finally, the assigned coding complexity dictates different fast partitioning strategies for each CU.

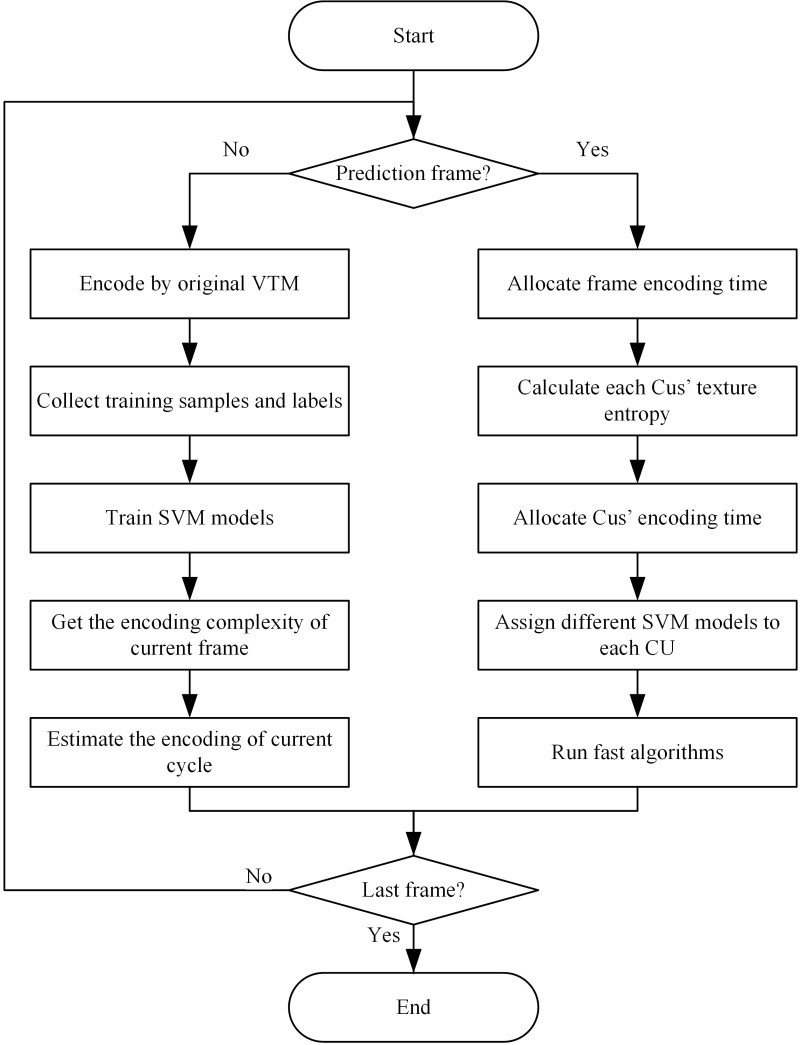

**Figure 1.** Flow chart of complexity control algorithm based on texture information entropy.

### 2.2. Frame-Level Complexity Estimation Algorithm

This paper aims to realize the video coding complexity control on the VVC platform. In all configured encoding processes, the encoding time of a specific frame cannot be accurately obtained in advance. In order to explore the relationship between the computational complexity of each frame, this paper selects MarketPlace, BasketballDrill, and Basket-

ballPass coded by the VVC test platform 3.0 (VVC Test Model, VTM) [30] in full intra configuration. The encoding time of each frame is counted when QP is 22, 27, 32, and 37. Table 1 shows the time taken for each sequence to encode frames 0 to 2, 9, 19, and 29. It indicates that non-adjacent frames have different computational complexities. For example, the coding time of the 9th frame and the 19th frame is quite different, so different complexity allocation strategies need to be adopted for different frames.. Since the 0th to 2nd frames have the characteristics of temporal correlation between adjacent frames, the difference in coding time is not obvious. Taking advantage of this feature, the proposed method sets 10 frames as a cycle and estimates the encoding time $T_{ori}$ of the entire cycle through the encoding time of the first frame of each cycle. The expression is as follows:

$$T_{ori} = T_n \times 10 \quad if \quad n\%10 = 0,$$ (1)

where $T_n$ represents the actual encoding time of the $n$th frame.

The target encoding time of other frames in the cycle will be obtained from the remaining target time and the number of remaining frames to be encoded, namely:

$$T_j = \frac{\beta \cdot T_{ori} - T_{encode}}{10 - g_{coded}},$$ (2)

where $T_j$ represents the target encoding time of $j$th frame in a cycle, $\beta$ represents the target complexity rate, $\beta \cdot T_{ori}$ represents the target encoding time of the entire sequence, $T_{encoded}$ represents the consumed encoding time, and $g_{coded}$ represents the number of completed encoded frames.

**Table 1.** Statistics of the encoding time of some frames (sec).

| Sequence | QP | POC | | | | | |
|---|---|---|---|---|---|---|---|
| | | 0 | 1 | 2 | 9 | 19 | 29 |
| MarketPlace | 22 | 410.27 | 407.26 | 402.19 | 417.11 | 371.31 | 400.38 |
| | 27 | 214.69 | 220.28 | 216.61 | 237.00 | 185.49 | 202.13 |
| | 32 | 121.40 | 124.78 | 124.17 | 148.20 | 102.41 | 111.67 |
| | 37 | 57.71 | 59.42 | 57.16 | 66.28 | 39.94 | 52.50 |
| BasketballDrill | 22 | 78.11 | 78.98 | 78.27 | 77.99 | 80.65 | 78.88 |
| | 27 | 59.59 | 61.27 | 60.23 | 59.03 | 62.05 | 64.17 |
| | 32 | 41.85 | 42.09 | 43.53 | 43.98 | 43.30 | 45.78 |
| | 37 | 23.92 | 22.79 | 24.30 | 23.14 | 24.22 | 24.10 |
| BasketballPass | 22 | 24.25 | 23.54 | 23.92 | 18.94 | 20.89 | 23.32 |
| | 27 | 17.99 | 18.18 | 18.14 | 13.40 | 15.96 | 17.23 |
| | 32 | 12.62 | 12.46 | 12.45 | 9.43 | 11.55 | 13.88 |
| | 37 | 7.82 | 7.38 | 7.80 | 5.65 | 6.77 | 9.51 |

The proposed method uses a frame-level complexity allocation strategy to retain good RD performance and more exact implementation complexity control. Normal encoding is carried out in time to prevent further deterioration of RD performance if the total of the actual time savings of the encoded frames exceeds the intended time savings of the entire sequence. Low complexity encoding will be used when the total of the actual time savings of the encoded frames is less than the desired time savings of the full sequence.

In VVC, there are six partition modes, those are QuadTree (QT) partition, Vertical Binary Tree (VB) partition, Horizontal Binary Tree (HB) partition, Vertical Ternary Tree (VT) partition, Horizontal Ternary Tree (HT) partition, and Non-Partitioning (NP). Multi-Type Tree (MT) Partitions refer to HB, VB, HT, and VT. One of the key components that contribute to VVC's increased compression effectiveness is its flexible partitioning structure. Each frame is split into numerous CTUs with a measurement of $128 \times 128$ in VVC coding. The CTUs are firstly divided by QT and then divided by QTMT at each QT leaf node. The RDO search is a top-down brute force check to ascertain the ideal coding depth of CUs in

each CTU. Therefore, redundant RDO searches need to be eliminated while maintaining optimal RD performance.

### 2.3. Frame-Level Complexity Control Algorithm

However, if only the complexity of the assignment is known, it is still impossible to encode each sequence accurately at the expected complexity. This paper proposes a $64 \times 64$ CU complexity control algorithm. We efficiently allocate the frame-level complexity to each CU while avoiding the unnecessary depth of the RDO process in the CU. This paper selects five sequences of different texture complexity and different resolutions and counts the proportion of each partition mode of $32 \times 32$ CUs under different QPs. Analyzing the data in Table 2, we can see that the proportion of non-partition modes also increases as the image texture becomes flatter with increasing QP. When the QP is the same, the sequences with relatively flat textures such as Tango2 have a better-undivided tendency than those of sequences with complex textures such as BasketballPass. It indicates that homogeneous CUs are more likely to be undivided and regions with richer textures have greater division depths. Therefore, predicting the QTMT partition depth enables the complexity to be effectively controlled. In addition, the proportion of NP is much larger than that of other partitions, indicating that selecting some of the division modes in advance can also effectively control the complexity.

**Table 2.** CU division result distribution (%).

| Sequence | QP | NP | QT | HB | VB | HT | VT |
|---|---|---|---|---|---|---|---|
| Tango2 | 22 | 57.18 | 2.21 | 22.59 | 12.71 | 2.62 | 2.69 |
| | 27 | 71.39 | 0.94 | 14.89 | 9.73 | 1.40 | 1.64 |
| | 32 | 76.64 | 0.66 | 12.29 | 7.94 | 1.16 | 1.31 |
| | 37 | 81.26 | 0.49 | 9.88 | 6.50 | 0.86 | 1.01 |
| CatRobot | 22 | 53.73 | 7.90 | 19.18 | 12.41 | 2.65 | 4.13 |
| | 27 | 58.49 | 5.60 | 17.32 | 12.43 | 2.36 | 3.80 |
| | 32 | 61.43 | 3.47 | 16.71 | 12.57 | 2.28 | 3.54 |
| | 37 | 65.02 | 1.62 | 15.68 | 12.43 | 1.98 | 3.27 |
| MarketPlace | 22 | 54.01 | 4.69 | 24.58 | 10.72 | 3.72 | 2.29 |
| | 27 | 58.49 | 3.30 | 23.49 | 8.76 | 4.14 | 1.81 |
| | 32 | 62.78 | 2.30 | 21.66 | 8.13 | 3.60 | 1.53 |
| | 37 | 68.43 | 1.52 | 18.82 | 7.10 | 2.93 | 1.20 |
| BasketballDrill | 22 | 56.67 | 29.55 | 5.93 | 3.22 | 2.52 | 2.12 |
| | 27 | 48.95 | 20.08 | 14.15 | 8.32 | 4.45 | 4.05 |
| | 32 | 49.90 | 10.72 | 20.20 | 11.30 | 4.00 | 3.87 |
| | 37 | 56.27 | 6.31 | 19.69 | 11.36 | 3.44 | 2.93 |
| BasketballPass | 22 | 44.41 | 14.45 | 21.04 | 8.27 | 7.48 | 4.34 |
| | 27 | 44.69 | 12.57 | 22.47 | 8.89 | 6.82 | 4.56 |
| | 32 | 47.19 | 10.69 | 23.02 | 8.57 | 6.43 | 4.10 |
| | 37 | 50.02 | 7.33 | 22.40 | 10.11 | 5.86 | 4.29 |

Information entropy can be used to measure the orderliness of the system. In image processing, it can measure the complexity of regions. The entropy corresponding to the image increases with the complexity of the image texture. Conversely, a simple image has less information and it has less entropy [31]. Therefore, this paper uses the texture entropy of the $i$th CU ($\omega_{(k,i)}$) in $k$th frame as the complexity weight, calculated by:

$$\omega_{(k,i)} = -\sum_{i=1}^{w'} \sum_{j=1}^{h'} d(i,j) \log(d(i,j)), \tag{3}$$

where the width and height of the corresponding CU are denoted as $w'$ and $h'$, and the probability distribution of the pixel value at $(i,j)$ is denoted by $d(i,j)$.

Then, the estimated coding time of the $i$th CU of the $k$th frame is $T_{(k,i)}$, namely:

$$T_{(k,i)} = T_k \times \frac{\omega_{(k,i)}}{\sum \omega_{(k,m)}}, \tag{4}$$

where $\sum \omega_{(k,m)}$ is the sum of all uncoded $64 \times 64$ CU complexity assignment weights in the $k$th frame. Then, the proposed method obtains the control threshold of $i$th $64 \times 64$ in the $k$th frame ($\varphi_{(k,i)}$), by:

$$\varphi_{(k,i)} = 10m \cdot T_{(k,i)} / T_{ori}, \tag{5}$$

where $m$ is the number of all $64 \times 64$ CUs in the $k$th frame.

In our algorithm, the partition direction of the CU is predicted by the SVM model. The Haar wavelet transform coefficients, texture entropy, and image energy are introduced [32]. There, the Haar wavelet transform coefficients can represent texture direction information. The image energy (ENG) can measure the uniformity of the pixel. The ENG is expressed as:

$$ENG = \sum_{i=1}^{w'} \sum_{j=1}^{h'} d(i,j)^2. \tag{6}$$

Besides, the SVM prediction algorithm uses online training. The training cycle is 10 frames. The training frame is placed on the first frame, and prediction frames are placed on the rest frames in one cycle. To effectively control encoding performance and time, the minimum number of partitions should be kept while preserving the optimal mode as much as possible. In this paper, statistical experiments are carried out on the performance of each fast SVM model. The training size and prediction size of CU are set to $64 \times 64$, $32 \times 32$, $32 \times 16$, $16 \times 32$, $16 \times 16$, $16 \times 8$, $8 \times 16$ and $8 \times 8$ pixels. However, in the actual test process, if the prediction model for $64 \times 64$ CU is introduced, the coding performance will decrease and the bitrate will increase a lot. The time spent training the $8 \times 8$ CU model exceeds the time it can save, so the six SVM models are finally retained. In this paper, the classifiers of six sizes are defined as $S_{32 \times 32}$, $S_{16 \times 16}$, $S_{32 \times 16}$, $S_{16 \times 32}$, $S_{8 \times 16}$, and $S_{16 \times 8}$.

In this paper, SVM models of different CU sizes are combined while the encoding performance of different combinations is tested and compared. The time saving rate is used to measure the complexity reduction performance calculated by the ratio of the encoding time obtained using the pattern to the encoding time of the original platform. Finally, four classifier combinations are retained named Patterns 1, 2, 3, and 4. Specifically, Patterns 1 to 4 are $\{S_{32 \times 16}, S_{16 \times 32}\}$, $\{S_{32 \times 32}, S_{8 \times 16}, S_{16 \times 8}\}$, $\{S_{32 \times 32}, S_{16 \times 16}\}$, and $\{S_{32 \times 32}, S_{32 \times 16}, S_{16 \times 32}, S_{16 \times 16}, S_{8 \times 16}, S_{16 \times 8}\}$, respectively. In Figure 2, the time saving ratio of four patterns is shown in turn from left to right. Each pattern is tested with three sequences. The three bars are MarketPlace in blue, BasketballDrill in red, and Basketball-Pass in green. The classifiers with the best time saving ratio performance are the two classifiers for rectangular CUs, $S_{32 \times 32}$ and $S_{16 \times 16}$, which are better than the classifiers for non-rectangular CUs. $S_{32 \times 32}$ and $S_{16 \times 16}$ can control encoding time between 65% and 75%. Therefore, this combination is used when $\varphi_{(k,i)}$ is $\in [0.65, 0.75)$. Combinations of non-rectangular CU classifiers will become applicable when the desired control rate decreases. During testing, the performance of classifiers for large-size CUs is better than that for small-size CUs, such as the combination of $S_{32 \times 16}$ and $S_{16 \times 32}$, which can save up more encoding time than the combination of $S_{8 \times 16}$ and $S_{16 \times 8}$. The time saving ratio of $S_{32 \times 16}$ and $S_{16 \times 32}$ is nearly 20% while that of $S_{16 \times 8}$ and $S_{8 \times 16}$ is relatively small. Therefore, Pattern 2 is used when $\varphi_{(k,i)}$ is $\in [0.85, \infty)$. For the pattern with $\varphi_{(k,i)} \in [0.75, 0.85)$, during the algorithm design phase, two classifier combinations are obtained by compromising the above two patterns, namely $\{S_{32 \times 32}, S_{8 \times 16}, S_{16 \times 8}\}$ and $\{S_{16 \times 16}, S_{32 \times 16}, S_{16 \times 32}\}$. After testing, $\{S_{32 \times 32}, S_{8 \times 16}, S_{16 \times 8}\}$ is set as Pattern 3 because it can achieve a 20% to 30% time saving ratio rather than 25% to 30% with the other combination, which is more in line with the expected control rate. When the six classifiers are fully on, this represents the time-saving boundary of our algorithm. Furthermore, the time saving ratio can reach 50%.

Therefore, using all classifiers is set as Pattern 4 with $\varphi_{(k,i)} \in [0, 0.65]$. Finally, the overall mapping relationship between $\varphi_{(k,i)}$ and Patterns is listed in Table 3.

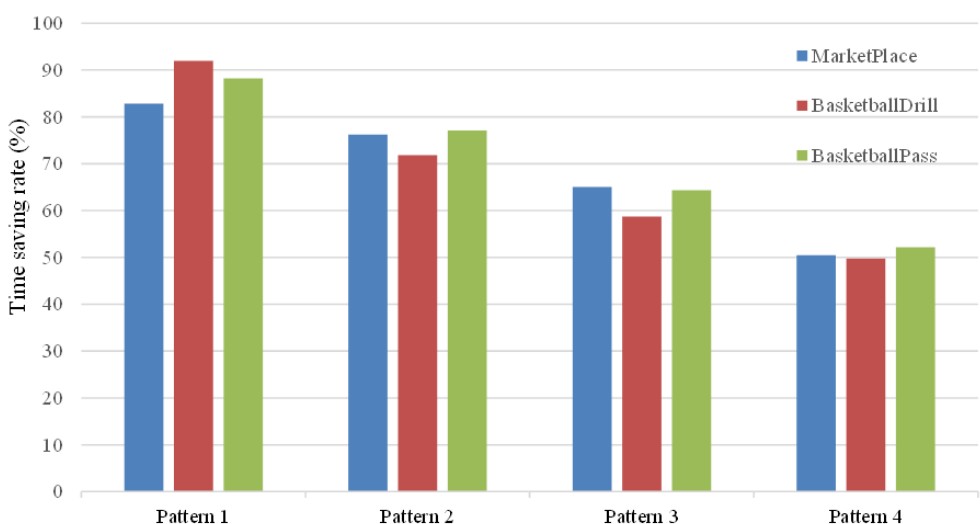

**Figure 2.** Classifier performance.

**Table 3.** Fast CU partition method.

| $\boldsymbol{\varphi}_{(k,i)}$ | Pattern |
| --- | --- |
| $[0.85, \infty)$ | $S_{32 \times 16}, S_{16 \times 32}$ |
| $[0.75, 0.85)$ | $S_{32 \times 32}, S_{8 \times 16}, S_{16 \times 8}$ |
| $[0.65, 0.75)$ | $S_{32 \times 32}, S_{16 \times 16}$ |
| $[0, 0.65)$ | $S_{32 \times 32}, S_{32 \times 16}, S_{16 \times 32}, S_{16 \times 16}, S_{8 \times 16}, S_{16 \times 8}$ |

## 3. Results

The method is practiced on the VVC test platform V TM3.0, using the all intra configuration. The test platform is 64G memory, Intel Core i9-10900XCPU 3.70 GHz with a 64-bit operating system. Bjntegaard Delta Bitrate (BDBR) [33] is used to evaluate the RD performance. *TS* is used to measure the actual time saving, calculated by:

$$TS = \frac{T_{Original} - T_{Proposed}}{T_{Original}} \times 100\%, \tag{7}$$

where $T_{Original}$ denotes the encoding time of the original VVC method, and $T_{Proposed}$ represents the actual encoding time of the proposed algorithm after setting the control precision. The control accuracy of the proposed algorithm is measured by the mean control error (*MCE*):

$$MCE = \frac{1}{4} \sum_{i=1}^{4} |TS_{QP_i} - (1 - \beta)|, \tag{8}$$

where $\beta$ represents the setting coding time ratio, and $TS_{QP_i}$ is the *TS* of the test sequence under $i$th *QP*. The *QP* set is $\{22, 27, 32, 37\}$.

Test and analysis are carried out for the cases where $\beta$ is 90%, 80%, 70%, and 60% respectively. Besides, the performance comparison with the state-of-the-art VVC intra complexity control method is performed.

Table 4 lists the complexity control performance of our method and the comparison method under different target complexities, in which the experiment result of [29] is marked with "*". Experiments show that the *MCE* of this method is small, which indicates that the time savings achieved at different QPs are not significantly different from each other. The average *TS* is close to $\beta$, and even slightly exceeds it. This shows that our complexity

control method has a more accurate and stable performance, is reasonable in different situations, and can adapt to reserving a small part of the complexity to cope with the sudden computing power demand in some mobile terminals. Therefore, the complexity control method of ours can meet the different computational capabilities.

**Table 4.** Comparison of coding performance under different target complexity (%). (*) method of [29].

| Sequence (Class) | $\beta = 90\%$ $\Delta BR/TS$ | $\beta = 80\%$ $\Delta BR/TS$ | $\beta = 70\%$ $\Delta BR/TS$ |
|---|---|---|---|
| Tango2 (A1) | 0.37/13.83 | 0.79/20.97 | 0.84/29.09 |
| FoodMarket4 (A1) | 0.23/3.78 | 0.60/9.58 | 0.64/13.95 |
| Campfire (A1) | 0.31/10.86 | 0.55/23.22 | 0.63/31.59 |
| CatRobot (A1) | 0.31/10.85 | 0.83/21.50 | 1.04/30.24 |
| DaylightRoad2 (A2) | 0.53/10.84 | 0.97/26.55 | 1.16/35.95 |
| ParkRunning3 (A2) | 0.61/14.01 | 0.38/17.94 | 0.49/25.20 |
| MarketPlace (B) | 0.20/11.15 | 0.56/19.47 | 0.66/31.39 |
| RitualDance (B) | 0.73/11.11 | 1.23 /24.62 | 1.46/36.62 |
| Cactus (B) | 0.34/10.63 | 0.69 /26.05 | 0.88/35.64 |
| BasketballDrive (B) | 0.40/14.68 | 0.80 /26.96 | 0.96/34.98 |
| BQTerrace (B) | 0.18/7.79 | 0.72 /27.78 | 0.85/37.57 |
| BasketballDrill (C) | 0.37/8.14 | 1.22 /23.18 | 1.64/38.26 |
| BQMall (C) | 0.32/9.63 | 0.90 /26.00 | 1.01/38.09 |
| PartyScene (C) | −0.01/4.77 | 0.46 /28.36 | 0.48/39.22 |
| RaceHorsesC (C) | 0.27/9.04 | 0.66 /27.43 | 0.77/37.09 |
| BasketballPass (D) | 0.50/10.60 | 1.09 /19.27 | 1.07/33.88 |
| BQSquare (D) | 0.00/13.42 | 0.45/ 18.86 | 0.52/35.81 |
| Blowing Bubbles (D) | 0.23/7.55 | 0.74 /26.32 | 0.75/37.47 |
| RaceHorsesD (D) | 0.15/7.64 | 0.66 /25.35 | 0.64/35.82 |
| Average | 0.32/10.01 | 0.75/23.13 | 0.87/33.57 |
| *MCE* | 2.30 | 4.69 | 5.86 |
| Average * | 0.23/9.96 | 0.44/17.72 | 0.66/25.55 |
| *MCE* * | 2.67 | 2.27 | 2.13 |

It can be observed that, at 90% of the expected complexity, the *TS* of different sequences is between 3.78% and 14.68%, with an average of 10.01%, and an *MCE* of 2.30. When the expected complexity is set to 80%, the *TS* of different sequences ranges from 9.58% to 22.14%, the average is 23.13%, and the *MCE* is 4.69. When the expected complexity is set to 70%, the *TS* of different sequences ranges from 13.95% to 38.09%, the average is 33.57%, and the *MCE* is 5.86. It is worth noting that a large gap with the $\beta$ comes from the FoodMarket4 sequence of class A1, resulting in less than ideal coding complexity control. This is because the sequence is more inclined to be divided into large blocks.

Table 4 also provides the RD performance of the method. These results show evidence that our method is robust to different settings when the sequence is encoded with different resolutions and different QPs. When the expected complexity was 90%, the BDBR of different sequences ranged from 0.18% to 0.61%, with an average of 0.32%. When the expected complexity is 80%, the BDBR of different sequences ranges from 0.13% to 0.72%, with an average of 0.36%. When the expected complexity was 70%, the BDBR of different sequences ranged from 0.49% to 1.64%, with an average of 0.87%. The BDBR increases with the control complexity, which is acceptable since the coding complexity and the BDBR are mutually constrained.

The proposed algorithm can save more time than the algorithm in [29] with the same target complexity. The average time saving of the algorithm in this paper exceeds the target complexity, while that in [29] is smaller than the target complexity. In contrast, this indicates that our algorithm is less prone to delays in encoding. It is worth mentioning that interactivity is very important for video conferencing, and the delay caused by encoding will seriously affect real-time interactivity. In these cases, saving more time rather than

saving more bitrate compared to the encoding time rate is a better solution for processing the video.

## 4. Conclusions

In this paper, we proposed a VVC intra complexity control algorithm with an application of texture entropy, which is able to accurately control the coding complexity under the condition of slight loss of RD performance, so that the VVC encoder can be used in different computing power and power-limited video equipment. The method firstly proposes a frame-level complexity allocation and control algorithm based on the phenomenon that the coding time ratio of adjacent frames is basically the same. Then, the proposed method uses the texture entropy and the CU decision model of SVM to adaptively control the coding complexity of the CU level. The experimental results show that the algorithm can effectively control the complexity when BDBR performs well and save more coding complexity than a state-of-the-art method, resulting in less delay. In the feature, we plan to control the inter-coding complexity on VVC with a learning method and features reflecting temporal correlation.

**Author Contributions:** Z.S. designed and completed the completed the algorithm and drafted the manuscript. J.L. co-designed the algorithm and polished the manuscript. Z.P. proofread the manuscript. F.C. analyzed the experimental results. M.Y. polished the manuscript. All authors have read and agreed to the published version of the manuscript.

**Funding:** This research was funded by the Natural Science Foundation of Chongqing under Grant No. cstc2021jcyj-msxmX0411, the Natural Science Foundation of Zhejiang Province under Grant No. LY20F010005, Science and Technology Research Program of Chongqing Municipal Education Commission under Grant No. KJZD-K202001105, and Scientific Research Foundation of Chongqing University of Technology under Grant Nos. 2020zdz029 and 2020zdz030.

**Conflicts of Interest:** The authors declare no conflict of interest.

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
