# Peer review of "Intra Complexity Control Algorithm for VVC"

_electronics, doi:10.3390/electronics11162572_

Round 1

Reviewer 1 Report

1. How does your method differ from other methods? Do a literature review of methods that are similar to yours. What are the disadvantages of other methods?

2. In which systems have you tested the proposed method?

3. How do you see the development of your method? What are the prospects for development?

4. In Table 2, you give the data of QP, NP, QT, BN, BV, TH, and TV, it is not clear what it is?

5. Fig. 2 What is marked in blue, red, and green? What are pattern1, pattern2, pattern3, and pattern4?

6. Conclusions are concise. Expand them.

Author Response

To Reviewer #1

  1. How does your method differ from other methods? Do a literature review of methods that are similar to yours. What are the disadvantages of other methods?

In original manuscript, we discussed and analyzed the VVC fast coding and HEVC complexity control. So far, Most of the computational complexity control algorithms are for HEVC, and few literatures are reported about the complexity control in VVC. Because of the different encoding techniques of HEVC and VVC, the complexity control algorithms for HEVC cannot be directly used in VVC. In the introduction of the revised manuscript, we have added a brief review of VVC complexity control. The computational control agoroithm proposed by Huang et. al. [1] is discussed. We also compared the proposed algorithm with Huang’s algorithm in the experimental section.

  1. In which systems have you tested the proposed method?

The test platform is 64G memory, Intel Core i9-10900XCPU 3.70 GHz with a 64-bit operating system. The detailed information have been added in the revised manuscript.

  1. How do you see the development of your method? What are the prospects for development?

Few literatures are reported about the complexity control in VVC. The proposed method explores the encoding process of VVC, and achieves the different computational complexity target. The proposed method can utilized in the real time video transmission scenario that the mobile terminal has different computing capabilities. Consequently, the video can be encoded and transmitted in time.

  1. In Table 2, you give the data of QP, NP, QT, BH, BV, TH, and TV, it is not clear what it is?

We apologize for our carelessness. Abbreviations in the table are inconsistent with definitions in the text. We have corrected the mistake in Table 2.

There are detailed technical descriptions of several abbreviations mentioned above. QP is the video quantization parameter. The larger the QP, the blurrier the video, and the smaller the QP, the clearer the video. The QT partition divides the current square CU equally into 4 sub-square CUs at its center point. The HB partition divides the current rectangular CU into two rectangular sub-CUs with a horizontal axis of symmetry. The VB division divides the current rectangular CU into two rectangular sub-CUs with a vertical axis of symmetry. The HT partition will horizontally divide the current rectangular CU into three rectangular sub-CUs with a ratio of 1:2:1. The VT division vertically divides the current rectangular CU into three rectangular sub-CUs at a ratio of 1:2:1. Finally, the encoder will perform RDO calculation on each sub-CU.

  1. 2 What is marked in blue, red, and green? What are pattern1, pattern2, pattern3, and pattern4?

Thanks for your suggestions. Pattern 1 to 4 are {S32×16, S16×32}, {S32×32, S8×16, S16×8}, {S32×32, S16×16}, and {S32×32, S32×16, S16×32, S16×16, S8×16, S16×8},respectively. In Fig. 2, the blue, red and green bars are time saving rate of MarketPlace, BasketballDrill, and BasketballPass, respectively.

  1. Conclusions are concise. Expand them.

       Thanks for the review of Conclusions. We have expanded. Here is the new one. In this paper, we proposed a VVC intra complexity control algorithm with an application of texture entropy, which is able to accurately control the coding complexity under the condition of slight loss of RD performance, so that the VVC encoder can be used in different computing power and power-limited video equipment. The method firstly proposes a frame-level complexity allocation and control algorithm based on the phenomenon that the coding time ratio of adjacent frames is basically the same. Then, the proposed method uses the texture entropy and the CU decision model of SVM to adaptively control the coding complexity of the CU level. The experimental results show that the algorithm can effectively control the complexity when BDBR performs well and save more coding complexity than a state-of-the-art method, resulting in less delay. In the feature, we plan to control inter-coding complexity on VVC by learning method and features reflecting temporal correlation.

References

[1] Huang Y, Xu J, Zhang L, et al. Intra Encoding Complexity Control with a Time-Cost Model for Versatile Video Coding[J]. arXiv preprint arXiv:2206.05889, 2022.

[2] Huang C; Peng Z; Xu Y; et al. Online learning-based multi-stage complexity control for live video coding. IEEE Transactions on Image Processing 2020, 30, 641-656.

[3] Cai Q, Chen Z, Wu DO, et al. Real-time constant objective quality video coding strategy in high efficiency video coding[J]. IEEE Transactions on Circuits and Systems for Video Technology, 2019, 30(7): 2215-2228.

[4] Li T; Xu M; Deng X; et al. Accelerate CTU partition to real time for HEVC encoding with complexity control. IEEE Transactions on Image Processing 2020, 29, 7482-7496.

Reviewer 2 Report

Please find the detail comments in the attached file.

Author Response

Dear editors and reviewers,

Thank you for your professional comments and feedback on our manuscript titled “Intra Complexity Control Algorithm for VVC”. These comments help us improve our manuscript and provide important guidance for future research. We have made extensive revisions for our previous manuscript based on the reviewers’ comments. We hope this meets your requirements for publication. In this revised pdf version, added changes to our manuscript are highlighted in yellow. The comments and regarding responses are detailed as follows.

To Reviewer #2

  1. It is obvious that the encoding time is different for each frame. However, it seems that it is not clear that the difference in the encoding time of each frame increases as the POC distance increase.
  2. There seems to be insufficient evidence to propose the method in this paper.

The purpose of the experiment is to show that the encoding time of adjacent frames is basically similar, and the encoding time of distant frames is different. Therefore, only a few subsequent frames can be estimated by encoding a frame normally.

It's not intended to state that the farther the POC is, the greater the difference in encoding time.

  1. Section Methods – Eq. (2)
  2. β represents the target complexity rate.
  3. Is it the same with Tc in Section 3. Results – line 234. If they are the same, need to match the notations, otherwise need an explanation for each notation.

Yes, it is the same. We have changed in the new manuscript.

  1. Section Methods – Figure 2.
  2. Need to indicate what the color of the bar graph means

We have added the description into the new manuscript. In Fig. 2, the blue, red and green bars are time saving rate of MarketPlace, BasketballDrill, and BasketballPass, respectively.

  1. The meaning of time saving is not clear
  2. Explain how to save time according to each pattern

Thanks for the suggestion. We have added the meaning of time saving in the manuscript.

In this paper, the time saving rate is used to measure the coding performance. The time saving rate is calculated by the ratio of the encoding time obtained using pattern to the encoding time of the original platform.

The Si×j means the fast SVM predicton model of i×j CU will be used. The SVM prediction model will predict the CU whether to be partitioned in horizontal or vertical direction. After the prediction, the number of sub-CU will decrease, leading to less RDO process, which is the most time consuming of VVC encoder.

  1. Section Methods – line 206
  2. Explain how each pattern group was formed

The patterns are obtained based on statistical experiments. Early in the algorithm design, we ran many combinations of SVMs. In the end, the pattern presented is the closest to the control scale, and is whose BDBR performs well.

  1. Section Methods – line 218
  2. When φ(k, i) [0.75, 0.85), two classifier combinations are obtained. But Table 3 shows only one combination.

Both combinations are tested during the algorithm design phase. In the end only one combination was determined.

We have rephrased it for clarity in the manuscript.

  1. Section Results
  2. According to CTC (Common Test Conditions) of VVC, when using all intra configuration, all-intra encoding is performed every 8 frames. Thus
  3. According to the 10-frame cycle proposed in this paper, frames with different similarities may belong to 10 frames.

As the reviewer said, all intra configuration is performed every 8 frames. The frames mentioned in the article are all down-sampled frames. For example, the 0th frame mentioned in the article is the 0th frame of the original sequence, the 1st frame mentioned in the article is the 8th frame of the original sequence, and so on.

  1. As the motive of the proposed method, the frame to which the fast algorithms is applied must be similar to the initial training frame to have a better effect.

Since the proposed method depends on the encoding time rate setting, it does not seem to have a significant effect, but it seems that an experiment suitable for the purpose is required.

Similar reaserch [1]-[4] perform result experiments like our paper, too. The encoding time rate is artificially set. Every 10% drop in encoding time rate requires a test of the algorithm.

  1. In order to evaluate the performance of the proposed method, it is necessary to compare

it with other methods.

       Thanks for your suggestion. We have added the comparison in our muanuscript. The comparative literature is [1], shown in Table 4.

Table 4. The coding performance of the algorithm under different target complexity

Sequence(class)

β=90%

β=80%

β=70%

∆ BR/TS

∆ BR/TS

∆ BR/TS

Tango2(A1)

0.37/13.83

0.79/20.97

0.84/29.09

FoodMarket4(A1)

0.23/3.78

0.60/9.58

0.64/13.95

Campfire(A1)

0.31/10.86

0.55/23.22

0.63/31.59

CatRobot(A1)

0.31/10.85

0.83/21.50

1.04/30.24

DaylightRoad2(A2)

0.53/10.84

0.97/26.55

1.16/35.95

ParkRunning3(A2)

0.61/14.01

0.38/17.94

0.49/25.20

MarketPlace(B)

0.20/11.15

0.56/19.47

0.66/31.39

RitualDance(B)

0.73/11.11

1.23 /24.62

1.46/36.62

Cactus(B)

0.34/10.63

0.69 /26.05

0.88/35.64

BasketballDrive(B)

0.40/14.68

0.80 /26.96

0.96/34.98

BQTerrace(B)

0.18/7.79

0.72 /27.78

0.85/37.57

BasketballDrill(C)

0.37/8.14

1.22 /23.18

1.64/38.26

BQMall(C)

0.32/9.63

0.90 /26.00

1.01/38.09

PartyScene(C)

-0.01/4.77

0.46 /28.36

0.48/39.22

RaceHorsesC(C)

0.27/9.04

0.66 /27.43

0.77/37.09

BasketballPass(D)

0.50/10.60

1.09 /19.27

1.07/33.88

BQSquare(D)

0.00/13.42

0.45/ 18.86

0.52/35.81

Blowing Bubbles (D)

0.23/7.55

0.74 /26.32

0.75/37.47

RaceHorsesD(D)

0.15/7.64

0.66 /25.35

0.64/35.82

Average

0.32/10.01

0.75/23.13

0.87/33.57

MCE

2.30

4.69

5.86

Average*

0.23/9.96

0.44/17.72

0.66/25.55

MCE*

2.67

2.27

2.13

References

[1] Huang Y, Xu J, Zhang L, et al. Intra Encoding Complexity Control with a Time-Cost Model for Versatile Video Coding[J]. arXiv preprint arXiv:2206.05889, 2022.

[2] Huang C; Peng Z; Xu Y; et al. Online learning-based multi-stage complexity control for live video coding. IEEE Transactions on Image Processing 2020, 30, 641-656.

[3] Cai Q, Chen Z, Wu DO, et al. Real-time constant objective quality video coding strategy in high efficiency video coding[J]. IEEE Transactions on Circuits and Systems for Video Technology, 2019, 30(7): 2215-2228.

[4] Li T; Xu M; Deng X; et al. Accelerate CTU partition to real time for HEVC encoding with complexity control. IEEE Transactions on Image Processing 2020, 29, 7482-7496.
